# Ionic Organic Solid 1,3-Bis(sulfomethyl)imidazoliumate as an Effective Metal-Free Catalyst for Sustainable Organic Syntheses

**DOI:** 10.3390/molecules28062695

**Published:** 2023-03-16

**Authors:** Mario Martos, Angélica M. Guapacha, Isidro M. Pastor

**Affiliations:** Organic Chemistry Department and Institute of Organic Synthesis (ISO), University of Alicante, ctra. San Vicente del Raspeig s/n, 03690 Alicante, San Vicente del Raspeig, Spain

**Keywords:** catalysis, imidazole, quinoline, sulfo-derivative, sustainable, synthetic methodology

## Abstract

The 1,3-bis(sulfomethyl)imidazole (bsmim) was effectively prepared by a multicomponent reaction, employing aminomethanesulfonic acid, glyoxal, and formaldehyde. The catalytic activity of bsmim was tested in the synthesis of quinoline derivatives, by means of the Friedländer reaction, and in the allylic substitution of alcohols with nitrogen-containing heterocycles. The performance of sulfo-imidazole derivative (bsmim) resulted in higher comparison with the carboxyimidazole analogs (bcmim and bcmimCl), under the same reaction conditions. This type of ionic organic solid allows the promotion of reactions in the absence of solvent and mild reaction conditions, which improves the sustainability of organic synthetic processes.

## 1. Introduction

Complementary to ionic liquids (ILs), an ionic solid formed by an organic cation and a counter-ion, with a melting point over 100 °C, constitutes an ionic organic solid (IOS). Similar to ILs, IOSs can be easily modulated via modification of the cation or anion, causing modifications in their properties [1,2]. Since its first reports in the mid-20th century [3,4], IOS research has developed in the fields of molecular electronics [3,5,6], optics [1,7,8,9,10], material science [11,12,13], and catalysis. In terms of catalysis, the scope of application of IOSs is diverse, due to the wide variety of these solids. They can be found in many organocatalytic processes, being of particular importance as photocatalysts [14,15], or asymmetric phase-transfer catalysts [16,17].

Imidazole-based IOSs are mainly substituted with aryl [18] and acyl [19] moieties, although some alkylimidazolium salts with highly branched or bulky side-chains have higher melting points [20]. The use of imidazolium IOSs in catalysis is mostly related to N-heterocyclic carbenes (NHC) both as ligands for metals [21,22] and as standalone NHC-catalysts [18], metal–organic frameworks [23,24], and Lewis/Brønsted acidic catalysis [25,26]. Regarding the acidic catalysis with imidazolium IOSs, the reports are related to the biscarboxy-functionalized imidazole derivatives. These imidazolic compounds constitute great heterogeneous catalysts due to their substantial Brønsted acidity [27], negligible solubility in aprotic and certain protic organic solvents [25], and high robustness [28]. Indeed, 1,3-bis(carboxymethyl)imidazolium chloride (bcmimCl) has been effectively applied as a promoter to a variety of synthetic protocols, including the preparation of quinolines and acridines via Friedländer reaction [25], the synthesis of dihydropyrimidones by the multicomponent Biginelli reaction [29], the obtention of 2,4-diarylthiophenes [30], and the allylation of anilines with allylic alcohols [28,31]. The latter is particularly interesting, as the regioselectivity of the process is controlled by the counter-anion of the imidazolium salt.

At this point, it can be postulated that the presence of sulfonic acid moieties could be of interest to improve the catalysis. Compared to carboxylic or hydrohalic acids, sulfonic acids are stronger by several order of magnitude [32,33], while being non-oxidizing (unlike comparably strong mineral acids such as perchloric acid) and having non-nucleophilic conjugate bases, thus, avoiding the side reactivity. Indeed, sulfo-imidazolium salts are well-known acidic catalysts, having been applied to a variety of transformations, including the synthesis of benzimidazoles via double condensation of benzaldehydes and 2-phenylenediamine [34], the preparation of coumarins by Pechmann reaction [35], or the depolymerization of lignocellulosic materials [36]. Based on these, the preparation of a sulfo-analogue of bcmim (i.e., 1,3-bis(sulfomethyl)imidazoliumate, bsmim) can be of interest as a second-generation Brønsted acidic imidazole IOS. 

The preparation of bis(sulfo)-functionalized imidazoles has been described by the ring opening of 1,4-butanesultone [37,38] and 1,3-propanesultone [39,40] using imidazole derivatives, with these sulfonated compounds being the most employed. Sodium 2-bromoethanesulfonate has been employed in the preparation of 1,3-bis(2-sulfoethyl)imidazole [41], but the analogue bis(sulfomethyl) has not been reported employing a similar procedure using halomethanesulfonates. Regarding the 1,3-bis(sulfomethyl)imidazoliumate (bsmim), there is only a patent by Armand and collaborators in which bsmim, among other sulfonates, is proposed as a component for solid electrolytes in lithium-ion batteries, but no further information regarding synthesis or properties is provided [42]. Therefore, a different approach based on a multicomponent reaction can be proposed, employing aminomethanesulfonic acid (AMSA) as depicted in Figure 1. Herein, we report our findings in the development of the multicomponent reaction to prepare a second-generation sulfo-imidazole acidic catalyst, and the study of its catalytic activity.

## 2. Results and Discussion

The preparation of 1,3-bis(sulfomethyl)imidazoliumate (bsmim, 1) was based in the experience of our research group for the synthesis of bis(carboxy)imidazoliumate derivatives, such as 1,3-bis(carboxymethyl)imidazoliumate (bcmim) [23,28] and 1,3-bis(3-carboxypropyl)imidazoliumate (bcpim) [43]. Thus, the synthetic plan was to construct the heterocycle by reaction of formaldehyde, glyoxal, and aminomethanesulfonic acid (AMSA) in a stoichiometric ratio 1:1:2 (Figure 1). Due to the very low solubility of AMSA in water, the use of the previously reported conditions [x6-8x] resulted in a sticky black tar from which no product could be isolated (Table 1, entry 1). As an attempt to mitigate this, additional water was added to the reaction (2 mL per 10 mmol of AMSA), observing full dissolution of the starting material within 20 min at 95 °C. After the reaction time elapsed, a brownish solid was isolated by solvent-induced precipitation in 45% yield (Table 1, entry 2). Proton NMR analysis confirmed the product bsmim (1), but also revealed the presence of unreacted AMSA and a significant amount of a by-product, which was identified as 1-methyl-3-(sulfomethyl)imidazoliumate [m(sm)im, 2] (Figure 1 and Table 1, entry 2). The formation of compound 2 can be rationalized by the decomposition of product 1 in the reaction medium (Figure 2).

The presence of compound **2** and the starting AMSA in the resulting mixture is problematic due to the need for several recrystallization steps, the yields being greatly diminished to half of the amount. Thus, preventing the formation of compound **2** was considered the best choice. The first approach assayed was the addition of excess carbonyl reactants (1.5 equivalents), resulting in the formation of a tar (Table 1, entry 3). Reducing the reaction time to 20 min allowed the obtention of **1** in 47% with a purity of 86% (Table 1, entry 4). The increase in the temperature with short time reaction resulted in the formation of more decomposition to the by-product **2** (Table 1, entries 5 and 6). According to the possible mechanism of the decomposition (Figure 2), it was considered that increasing the pH of the reaction media could prevent this pathway. Pure product **1** was obtained in the presence of a bit amount of sulfuric acid, the 30% yield proving its efficiency in preventing the decomposition (Table 1, compare entry1 and entry 7). In addition, the solubility of AMSA was increased in this acidic medium, so additional water (2 mL) afforded pure bsmim (**1**) in 49% yield (Table 1, entry 8). The product **1** evidenced clearly a C2 symmetry axis (C2 to the C4-C5 bond) by recording the ^1^H and ^13^C-NMR spectra. As expected, all H-chemical shifts (9.27, 7.77, and 5.43 ppm) are higher than those observed for the carboxy-analogue bcmim [23].

To evaluate the environmental impact of the synthetic procedure, sustainable metrics were calculated as compiled in Table 2. The E-factor is 18.5 (mass of waste per unit mass of product), for which the use of solvent for the purification step is the main factor contributing to this value. Still, the E-factor remains comfortably within the lower limits of fine chemical production (5 to 50) [44] and besides, even with a hefty 25 point penalty due to the yield obtained, the EcoScale score remains in the higher range of “acceptable” methodologies (50 to 75 points), showing the potential of the synthetic process [45]. In general terms, the synthetic process has acceptable scores, considering the difficulties encountered throughout the optimization studies. The high theoretical efficiency of this multicomponent imidazolium synthesis reflects on the excellent atom economy (AE, Table 2) and stoichiometric factor (SF, Table 2) values, which partially compensate for the moderate reaction yield (RY) and low reaction mass efficiency (RME) and material recovery parameter (MRP) values. In addition, the combination of the overall material efficiency parameters of RY, AE, 1/SF, MRP, and RME in a vector magnitude ratio (VMR, Table 2) can provide unbiased quantification of the overall degree of greenness. The VMR was calculated according to equation 1 [46], giving a value of 0.623 for the preparation of bsmim (**1**), which is satisfactory considering the easy decomposition of compound **1** in the reaction mixture. For comparison, green metrics for previous IOSs (i.e., bcmim and bcmimCl) were calculated (Table 2), obtaining VMR values similar to the calculated for bsmim. The slightly better VMR values for bcmim and bcmimCl are due to the higher reaction yields (almost double), which are related to the absence of decomposition of these IOSs during their preparation.
(1)$$VMR=\frac{1}{\sqrt{5}}\left[\sqrt{\;{\left(RY\right)}^{2}+{\left(AE\right)}^{2}+{\left(\frac{1}{SF}\right)}^{2}+{\left(MRP\right)}^{2}+{\left(RME\right)}^{2}}\right]$$

To assess the catalytic activity of the newly prepared bsmim (**1**) against the first-generation IOS based on biscarboxy-imidazole derivatives, the Friedländer synthesis of quinolines and the allylation of heterocycles with allylic alcohols were selected as comparative organic transformations. For the synthesis of quinolines, 2-aminobenzophenone was combined with pentane-2,4-dione in the presence of a substoichiometric amount of **1** (10 mol%) under neat conditions. After heating the reaction (80 °C) for 16 h, the quinoline **3** was obtained with quantitative yield (Figure 3). Remarkably, the carboxy analogue, 1,3-bis(carboxymethyl)imidazoliumate (bcmim) only achieved 9% conversion to the quinoline **3**, under the same working conditions [25], proving the stark difference in baseline catalytic activity between the sulfo and carboxy functionalized IOS. For the synthesis of quinoline **3**, it was observed that bsmim has a similar catalytic activity to the 1,3-bis(carboxymethyl)imidazolium chloride (bcmimCl), which is a more active derivative providing compound **3** with 94% yield (Figure 3). 

Next, a variety of different ketones were reacted with 2-aminobenzophenones in the presence of bsmim (**1**), employing the same reaction conditions as previously reported in the bcmimCl-catalyzed process. The yields of the products were compared with those reported when appropriate (Figure 3). Using bsmim as catalyst resulted in products being obtained in excellent-to-quantitative yields across the board. The protocol tolerates a variety of substrates, including acyclic and cyclic ketones, diketones, and keto esters (Figure 3). A slight reduction in yields is observed with electron-withdrawing substituents in the aminobenzophenone, as evidenced by compound **5** (Figure 3). Compared to bcmimCl, the sulfo derivative, bsmim, achieves up to 15% higher yields in this reaction (Figure 3), consistently outperforming the first-generation IOS. 

To expand the study of its catalytic activity, the reaction between allylic alcohols and nitrogen-containing heterocycles was carried out, forming the corresponding allylated heterocycles. It should be noted that the reaction between (*E*)-1,3-diphenylprop-2-en-1-ol and indole in the presence of the carboxy analog (bcmim) did not give rise to any product, the starting materials being recovered (Figure 4) [47]. In contrast, the use of the chloride derivative (bcmimCl) gave the expected product in 93% yield (Figure 4) [47]. At this point, the catalytic activity of bsmim (**1**) was tested under the same reaction conditions (i.e., 10 mol% of catalyst bsmim, neat, 80 °C, 2 h), isolating the allylindole **11** in 98% yield. This experiment proves the improved reactivity of the sulfo- vs. carboxy-imidazole derivative, being similar or slightly superior to the carboxy-imidazolium chloride. The allylation reaction was performed for representative substrates under the same reaction conditions, the results being compiled in Figure 4. The results obtained indicate that bsmim is superior to bcmimCl in terms of catalytic activity, as evidenced by the preparation of compound **12** (Figure 4). For the rest of compounds, yields are consistently in the excellent-to-quantitative range (Figure 4). 

As a final point, the stability of the bsmim (**1**) is compromised by prolonged heating, so the potential for recycling is reduced in the organic syntheses studied herein. Thus, in lieu of its higher activity, bsmim might be better suited for the design of milder synthetic processes, which would also better preserve its integrity.

## 3. Materials and Methods

All commercially available reagents were purchased (Acros, Aldrich, and Fluka) and used without further purification. Melting points were determined using a Gallenkamp capillary melting point apparatus (model MPD 350 BM 2.5) and are uncorrected. The 1H and 13C nuclear magnetic resonance (NMR) spectra were recorded (Appendix A) at the Research Technical Services of the University of Alicante, Alicante, Spain (SSTTI-UA; https://sstti.ua.es/en; accessed on 23 February 2023), employing a Bruker AC-300 (Madrid, Spain). Chemical shifts (δ) are given in ppm, and the coupling constants (*J*) are given in Hz. Deuterated chloroform (CDCl_3_) and water (D_2_O) were used as solvent. Low-resolution mass spectrum (LRMS) with electronic ionization (EI) was obtained at the Research Technical Services of the University of Alicante (SSTTI-UA; https://sstti.ua.es/en (accessed on 23 February 2023)) with an Agilent model 5973 Network mass spectrometer with a direct introduction of the sample to the ionic source using the SIS Direct Insertion Probe (73DIP-1). The mass spectrometer is equipped with a single electronic impact source as well as a quadrupole analyzer. Detected fragmentations are given as *m*/*z* with relative intensities in parenthesis (%). Elemental analysis (CHNS) was performed at the X-ray unit (SSTTI-UA; https://sstti.ua.es/en (accessed on 23 February 2023)), using the microanalyzer with Micro TruSpec detection system from LECO.

### 3.1. Procedure for the Preparation of Bsmim (1)

Glyoxal (40% aq., 0.57 mL, 5 mmol), formaldehyde (37% aq., 0.37 mL, 5 mmol) and aminomethanesulfonic acid (1.11 g, 10 mmol) were added to a round-bottom flask along with water (2 mL) and sulfuric acid (catalytic, 0.1 mL) and stirred at 95 °C for 20 min, after which it was quickly put in an ice–brine bath. Then, acetone (10 mL) was added, and the mixture was vigorously stirred (1200 rpm), which caused the precipitation of a fluffy brown solid over the next 20 min. The solid was then filtered, affording 628 mg of pure bsmim (49% yield).

1,3-Bis(sulfomethyl)imidazole (bsmim, **1**): light brown solid, 49% yield; m.p. = 291–294 °C; ^1^H NMR (400 MHz, D_2_O) δ_H_ = 9.26 (s, 1H, NCHN), 7.76 (d, *J* = 1.6 Hz, 2H, NCHCHN), 5.43 (s, 4H, CH_2_); ^13^C NMR (100 MHz, D_2_O) δ_C_ = 138.3, 123.5, 62.6. MS (EI, 70 eV) *m*/*z* (%): 66 (5), 64 (100), 48 (40); MS/MS (ESI^+^) [256]: 242 (26), 238 (20), 193 (13), 176 (100), 163 (27); elemental analysis calcd. for C_5_H_8_N_2_O_6_S_2_: C (23.4%), H (3.2%), N (10.9%), O (37.5%), S (25.0%); found: C (22.5%), H (3.7%), N (10.2%), O (39.3%), S (24.3%).

### 3.2. General Procedure for the Synthesis of Quinolines Promoted by Bsmim

2-Aminobenzophenone (0.5 mmol, 99 mg), ketone (2.5 mmol), and bsmim (10 mol%, 13 mg) were added to a reaction tube. The reaction was then stirred at 80 °C for 16 h, after which ethyl acetate was added (1 mL). After filtering the organic phase to remove the catalyst and removing the solvent under vacuum, the crude product was purified by column chromatography using mixtures of hexane and ethyl acetate.

3-Acetyl-2-methyl-4-phenylquinoline (**3**): yellow solid, purification by column chromatography (hexane/ethyl acetate 7:3), 99% yield; m.p. 120–122 °C (lit. 117–119 °C) [48]; ^1^H NMR (300 MHz, CDCl_3_): δ_H_ = 8.13 (d, *J* = 8.4 Hz, 1H, CH_Ar_), 7.74 (ddd, *J* = 8.4, 6.9, 1.4 Hz, 1H, CH_Ar_), 7.65–7.62 (m, 1H, CH_Ar_), 7.53–7.46 (m, 4H, CH_Ar_), 7.38–7.35 (m, 2H, CH_Ar_), 2.73 (s, 3H, NCCH_3_), 2.00 (s, 3H, COCH_3_); ^13^C NMR (100 MHz, CDCl_3_): δ_C_ = 205.5, 153.7, 147.1, 144.7, 135.2, 134.9, 130.6, 130.1, 129.2, 128.9, 128.6, 126.9, 126.4, 125.2, 32.1, 23.7; MS (EI, 70 eV) *m/z* (%): 261 (M^+^, 49), 247 (19), 246 (100), 219 (7), 218 (38), 217 (31), 176 (22), 151 (8).

3-(Ethoxycarbonyl)-2-methyl-4-phenylquinoline (**4**): yellow solid, purification by column chromatography (hexane/ethyl acetate 9:1), 98% yield; m.p. 99–100 °C (lit. 98–100 °C) [48]; ^1^H NMR (400 MHz, CDCl_3_): δ_H_ = 8.06 (d, *J* = 8.1 Hz, 1H, CH_Ar_), 7.68 (ddd, *J* = 8.4, 6.8, 1.4 Hz, 1H, CH_Ar_), 7.56 (dd, *J* = 8.4, 0.9 Hz, 1H, CH_Ar_), 7.47–7.43 (m, 3H, CH_Ar_), 7.41–7.33 (m, 3H, CH_Ar_), (q, *J* = 7.1 Hz, 2H, CH_2_), 2.78 (s, 3H, NCCH_3_), 0.92 (t, *J* = 7.1 Hz, 3H, COCCH_3_); ^13^C NMR (100 MHz, CDCl_3_): δ_C_ = 168.5, 154.6, 147.7, 146.4, 135.8, 130.3, 129.4, 128.9, 128.5, 128.3, 127.5, 126.5, 126.5, 125.2, 61.4, 23.8, 13.7; MS (EI, 70 eV) *m/z* (%): 292 (M^+^+1, 20), 291 (M^+^, 96), 263 (8), 262 (8), 247 (20), 246 (100), 245 (34), 219 (10), 218 (44), 217 (41), 216 (14), 177 (8), 176 (23), 151 (8).

3-Acetyl-6-chloro-2-methyl-4-phenylquinoline (**5**): yellow solid, purification by column chromatography (hexane/ethyl acetate 85:15), 93% yield; m.p. 160–161 °C (lit. 149–152 °C) [48]; ^1^H NMR (400 MHz, CDCl_3_): δ_H_ = 8.06 (d, *J* = 8.9 Hz, 1H, CH_Ar_), 7.66 (dd, *J* = 8.9, 2.3 Hz, 1H, CH_Ar_), 7.58 (d, *J* = 2.3 Hz, 1H, CH_Ar_) 7.55–7.52 (m, 3H, CH_Ar_), 7.35–7.32 (m, 2H, *J* = 8.9, 2.3 Hz, 1H, CH_Ar_),, 2.69 (s, 3H, NCCH_3_), 1.99 (s, 3H, COCH_3_); ^13^C NMR (100 MHz, CDCl_3_): δ_C_ = 204.9, 154.1, 145.3, 143.9, 135.7, 134.4, 132.9, 131.5, 130.0, 129.5, 129.1, 126.1, 125.1, 31.9, 23.6; MS (EI, 70 eV) *m*/*z* (%): 297 (M^+^+2, 16), 296 (M^+^+1, 10), 295 (M^+^, 46), 282 (33), 281 (19), 280 (100), 254 (7), 252 (22), 218 (8), 217 (32), 216 (13), 189 (8), 176 (23).

9-Phenyl-1,2,3,4-tetrahydroacridine (**6**): yellow solid, purification by column chromatography (hexane/ethyl acetate 8:2), 94% yield; m.p. 151–153 °C (lit. 152–155 °C) [25]; ^1^H NMR (300 MHz, CDCl_3_): δ_H_ = 8.13 (d, *J* = 8.5 Hz, 1H, CH_Ar_), 7.66–7.60 (m, 1H, CH_Ar_), 7.56–7.45 (m, 3H, CH_Ar_), 7.35–7.33 (m, 2H, CH_Ar_), 7.25–7.22 (m, 2H, CH_Ar_), 3.27 (t, *J* = 6.6 Hz, 2H, NCCH_2_), 2.62 (t, *J* = 6.4 Hz, 2H, PhCCCH_2_), 2.02–1.93 (m, 2H, NCCCH_2_), 1.84–1.76 (m, 2H, PhCCCCH_2_); ^13^C NMR (75 MHz, CDCl_3_): δ_C_ = 158.9, 147.7, 145.4, 136.9, 129.1, 129.0, 128.8, 128.7, 128.1, 127.7, 126.9, 126.0, 125.9, 33.8, 28.1, 23.0, 22.8; MS (EI, 70 eV) *m/z* (%): 260 (M^+^+1, 19), 259 (M^+^, 100), 258 (M^+^-1, 66), 244 (8), 230 (14), 182 (8).

9-Phenyl-2,3-dihydro-1*H*-cyclopenta[*b*]quinoline (**7**): faint yellow solid, purification by column chromatography (hexane/ethyl acetate 98:2), 93% yield, m.p. 138–141 °C (lit. 138–139 °C) [25]; ^1^H NMR (400 MHz, CDCl_3_): δ_H_ = 8.02–8.00 (m, 1H, CH_Ar_), 7.57–7.53 (m, 2H, CH_Ar_), 7.47–7.37 (m, 3H, CH_Ar_), 7.32–7.28 (m, 3H, CH_Ar_), 3.17 (t, *J* = 7.7 Hz, 2H, NCCH_2_), 2.83 (t, *J* = 7.4 Hz, 2H, PhCCCH_2_), 2.13–2.07 (m, 2H, NCCCH_2_); ^13^C NMR (100 MHz, CDCl_3_): δ_C_ = 167.5, 147.9, 142.9, 136.9, 133.8, 129.4, 128.9, 128.6, 128.4, 128.1, 126.4, 125.8, 125.7, 35.3, 30.5, 23.7; MS (EI, 70 eV) *m/z* (%): 246 (M^+^+1, 18), 245 (M^+^, 100), 244 (M^+^-1, 80), 243 (12), 242 (12), 241 (7), 217 (9), 168 (15), 167 (10), 108 (7).

12-Phenyl-6,7,8,9,10,11-hexahydrocycloocta[*b*]quinoline (**8**): faint yellow solid, purification by column chromatography (hexane/ethyl acetate 95:5), 96% yield; m.p. 209–211 °C (lit. 214–215 °C) [48]; ^1^H NMR (300 MHz, CDCl_3_): δ_H_ = 8.01–7.98 (m, 1H, CH_Ar_), 7.54–7.48 (m, 1H, CH_Ar_), 7.46–7.36 (m, 3H, CH_Ar_), 7.27–7.11 (m, 4H, CH_Ar_), 3.17–3.13 (m, 2H, NCCH_2_), 2.69–2.65 (m, 2H, PhCCCH_2_), 1.88–1.84 (m, 2H, CH_2_), 1.42–1.27 (m, 6H, 3xCH_2_); ^13^C NMR (75 MHz, CDCl_3_): δ_C_ = 163.5, 146.7, 146.3, 137.7, 131.9, 129.4, 128.5, 128.4, 127.7, 127.3, 126.2, 125.5, 36.3, 31.3, 31.3, 28.2, 26.8, 25.9; MS (EI, 70 eV) *m/z* (%): 288 (M^+^+1, 20), 287 (M^+^, 93), 286 (M^+^-1, 100), 272 (9), 260 (8), 259 (17), 258 (40), 256 (9), 245 (8), 244 (25), 243 (9), 242 (9), 232 (19), 231 (14), 230 (16), 217 (14), 216 (7), 202 (8), 189 (10).

9-Phenyl-3,4-dihydroacridin-1(2*H*)-one (**9**): yellow solid, purification by column chromatography (hexane/ethyl acetate 8:2), 96% yield; m.p. 163–165 °C (lit. 161–162 °C) [25]; ^1^H NMR (400 MHz, CDCl_3_): δ_H_ = 8.18 (d, *J* = 8.4 Hz, 1H, CH_Ar_), 7.80 (ddd, *J* = 8.4, 6.6, 1.6 Hz, 1H, CH_Ar_), 7.53–7.41 (m, 5H, CH_Ar_), 7.19–7.17 (m, 2H, CH_Ar_), 3.45 (t, *J* = 6.3 Hz, 2H, NCCH_2_), 2.72 (t, *J* = 6.1 Hz, 2H, COCH_2_), 2.28–2.25 (m, 2H, COCCH_2_); ^13^C NMR (100 MHz, CDCl_3_): δ_C_ = 197.6, 162.1, 152.5, 147.7, 137.4, 132.4, 128.4, 128.3, 128.1, 127.9, 127.7, 126.9, 124.0, 40.7, 34.1, 21.4; MS (EI, 70 eV) *m*/*z* (%): 274 (M^+^+1, 19), 273 (M^+^, 100), 272 (M^+^-1, 94), 246 (7), 245 (47), 244 (97), 217 (32), 216 (31), 214 (8), 190 (9), 189 (16), 176 (7).

2,9-Diphenyl-1,2,3,4-tetrahydroacridine (**10**): yellow solid, purification by column chromatography (hexane/ethyl acetate 9:1), 97% yield; ^1^H NMR (400 MHz, CDCl_3_): δ_H_ = 7.98–7.96 (m, 1H, CH_Ar_), 7.57–7.52 (m, 1H, CH_Ar_), 7.41–7.35 (m, 3H, CH_Ar_), 7.25–7.10 (m, 9H, CH_Ar_), 3.37–3.22 (m, 2H, CH_Alk_), 2.95 (tdd, *J* = 11.4, 4.7, 3.0 Hz, 1H, CH_Alk_), 2.83 (ddd, *J* = 17.1, 4.9, 2.0 Hz, 1H, CH_Alk_), 2.68 (dd, *J* = 17.1, 4.9 Hz, 1H, CH_Alk_), 2.25–2.19 (m, 1H, CH_Alk_), 2.13–2.02 (m, 1H, CH_Alk_); ^13^C NMR (100 MHz, CDCl_3_): δ_C_ = 158.4, 146.9, 146.6, 145.9, 136.9, 129.2, 129.0, 128.9, 128.8, 128.7, 128.7, 128.5, 127.9, 127.8, 127.0, 126.8, 126.5, 126.0, 125.7, 40.9, 36.2, 34.4, 30.3; MS (EI, 70 eV) *m/z* (%): 336 (M^+^ + 1, 14), 335 (M^+^, 65), 334 (M^+^ − 1, 100), 231 (8), 230 (19), 91 (8).

### 3.3. General Procedure for the Allylation of Heterocycles Promoted by Bsmim

In a glass tube, precisely weighed allyl alcohol (0.5 mmol), heterocycle (0.5 mmol), and bsmim (10 mol%) were added. The mixture was then stirred at 80 °C until completion (monitored by GC-MS), after which the crude reaction mixture was diluted with ethyl acetate (1 mL) and filtered through a thin plug of silica to remove the catalyst. After evaporation of the solvent under reduced pressure, the corresponding allyl heterocycles were obtained, which were purified by column chromatography or preparative TLC when required.

(*E*)-3-(1,3-Diphenylallyl)-1*H*-indole (**11**): yellow oil, obtained pure, 99% yield; ^1^H NMR (400 MHz, CDCl_3_): δ_H_ = 7.96 (br s, 1H, NH), 7.51 (d, *J* = 8.0 Hz, 1H, CH_Ar_), 7.45–7.22 (m, 12H, CH_Ar_), 7.10 (ddd, *J* = 8.0, 7.1, 1.0 Hz, 1H, CH_Ar_), 6.93 (d, *J* = 1.3 Hz, 1H, CH_Ar_), 6.80 (dd, *J* = 15.9, 7.4 Hz, 1H, PhC=CH), 6.52 (br d, *J* = 15.9 Hz, 1H, C=CHPh), 5.19 (br d, *J* = 7.4 Hz, 1H, C=CCH); ^13^C NMR (100 MHz, CDCl_3_): δ_C_ = 143.5, 137.6, 136.8, 132.7, 130.7, 128.6, 128.6, 127.3, 126.9, 126.5, 126.5, 122.7, 122.2, 120.0, 119.6, 118.8, 111.2, 46.3; MS (EI, 70 eV) *m/z* (%): 310 (M^+^ + 1, 25), 309 (M^+^, 100), 308 (M^+^ − 1, 39), 294 (10), 233 (7), 232 (36), 231 (7), 230 (20), 218 (15), 217 (17), 206 (28), 205 (8), 204 (22), 202 (8), 192 (14), 191 (16), 178 (9), 130 (19), 115 (17).

(*E*)-3-(1,3-Diphenylallyl)-2-phenyl-1*H*-indole (**12**): faint orange oil, purified by preparative TLC (hexane/ethyl acetate 9:1), 60% yield; ^1^H NMR (300 MHz, CDCl_3_): δ_H_ = 8.02 (br s, 1H, NH), 7.52–7.48 (m, 2H, CH_Ar_), 7.44–7.12 (m, 16H, CH_Ar_), 6.98 (ddd, *J* = 8.0, 7.1, 1.0 Hz, 1H, CH_Ar_), 6.88 (dd, *J* = 15.8, 7.3 Hz, 1H, PhC=CH), 6.39 (dd, *J* = 15.8, 1.0 Hz, 1H, C=CHPh), 5.27 (br d, *J* = 7.3 Hz, 1H, C=CCH); ^13^C NMR (75 MHz, CDCl_3_): δ_C_ = 143.6, 137.6, 136.3, 135.7, 133.0, 132.4, 131.2, 128.9, 128.7, 128.6, 128.4, 128.2, 128.0, 127.2, 126.4, 126.2, 122.2, 121.3, 119.8, 113.9, 111.1, 45.2; MS (EI, 70 eV) *m*/*z* (%): 386 (M^+^ + 1, 30), 385 (M^+^, 100), 384 (M^+^ − 1, 19), 309 (12), 308 (46), 307 (7), 306 (17), 304 (11), 295 (21), 294 (96), 293 (29), 292 (9), 291 (11), 282 (17), 281 (8), 280 (18), 278 (8), 230 (14), 218 (9), 217 (13), 207 (9), 206 (47), 205 (8), 204 (23), 203 (8), 202 (9), 194 (9), 193 (45), 192 (22), 191 (22), 189 (8), 178 (9), 176 (7), 165 (14), 153 (9), 152 (10), 146 (9), 115 (9), 91 (7).

(*E*)-3-(1,3-Diphenylallyl)-9-ethyl-1*H*-indole (**13**): reddish oil, obtained pure, 99% yield; ^1^H NMR (300 MHz, CDCl_3_) δ_H_ = 7.89 (br s, 1H, NH), 7.36–7.15 (m, 11H, CH_Ar_), 7.02–6.97 (m, 2H, CHN, CH_Ar_), 6.84 (dd, *J* = 2.4, 0.9 Hz, 1H, CH_Ar_), 6.72 (dd, *J* = 15.8, 7.4 Hz, 1H, PhC=CH), 6.43 (br d, *J* = 15.8 Hz, 1H, C=CHPh), 5.10 (br d, *J* = 7.4 Hz, 1H, C=CCH), 2.82 (q, *J* = 7.6 Hz, 2H, CH_2_), 1.34 (t, *J* = 7.6 Hz, 3H, CH_3_); ^13^C NMR (75 MHz, CDCl_3_): δ_C_ = 143.6, 137.6, 135.6, 132.7, 130.6, 128.6, 128.6, 128.5, 127.3, 126.7, 126.6, 126.4, 122.3, 120.7, 119.8, 119.2, 117.8, 46.4, 24.1, 13.9; MS (EI, 70 eV) *m*/*z* (%): 338 (M^+^ + 1, 27), 337 (M^+^, 100), 336 (M^+^ − 1, 34), 334 (9), 333 (28), 332 (9), 308 (20), 261 (7), 260 (33), 234 (23), 231 (9), 230 (24), 218 (9), 217 (14), 204 (10), 192 (17), 191 (20), 189 (7), 158 (15), 130 (12), 115 (11); HRMS (QTOF) calculated for C_25_H_23_N = 337.1830, observed = 337.1823.

(*E*)-1-(1,3-Diphenylallyl)-1*H*-pyrazole (**14**): colorless oil, obtained pure, 99% yield; ^1^H NMR (300 MHz, CDCl_3_): δ_H_ = 7.60 (d, *J* = 1.4 Hz, 1H, CH_Ar_), 7.45 (d, *J* = 2.2 Hz, 1H, CH_Ar_), 7.39–7.20 (m, 10H, CH_Ar_), 6.71 (dd, *J* = 16.0, 6.9 Hz, 1H, PhC=CH), 6.43 (br d, *J* = 16.0 Hz, 1H, C=CHPh), 6.30–6.28 (m, 1H, CH_Ar_), 6.18 (br d, *J* = 6.9 Hz, 1H, C=CCH); ^13^C NMR (75 MHz, CDCl_3_): δ_C_ = 139.6, 139.4, 136.1, 133.8, 128.9, 128.7, 128.2, 127.4, 127.3, 126.8, 105.7, 67.6; MS (EI, 70 eV) *m/z* (%): 261 (M^+^ + 1, 15), 260 (M^+^, 78), 259 (M^+^-1, 24), 232 (7), 193 (28), 192 (43), 191 (55), 190 (11), 189 (23), 184 (8), 183 (55), 179 (7), 178 (33), 169 (31), 165 (21), 157 (18), 156 (36), 144 (21), 143 (8), 117 (9), 115 (100), 91 (27), 89 (7), 77 (8).

(*E*)-1-(1,3-Diphenylallyl)-3,5-dimethyl-1*H*-pyrazole (**15**): white solid, obtained pure, 99% yield; m.p. 108–110 ºC; ^1^H NMR (400 MHz, CDCl_3_): δ_H_ = 7.43–7.41 (m, 2H, CH_Ar_), 7.33–7.23 (m, 6H, CH_Ar_), 7.16–7.14 (m, 2H, CH_Ar_), 6.88 (dd, *J* = 15.9, 7.5 Hz, 1H, PhC=CH), 6.51 (br d, *J* = 15.9 Hz, 1H, C=CHPh), 5.98 (br d, *J* = 7.5 Hz, 1H, C=CCH), 5.87 (s, 1H, MeCCH), 2.27 (s, 3H, CH_3_), 2.17 (s, 3H, CH_3_); ^13^C NMR (100 MHz, CDCl_3_): δ_C_ = 147.8, 140.2, 139.2, 136.5, 133.1, 128.8, 128.6, 128.0, 127.9, 127.7, 126.9, 126.9, 105.9, 64.5, 13.8, 11.5; MS (EI, 70 eV) *m/z* (%): 289 (M^+^+1, 10), 288 (M^+^, 50), 287 (M^+^-1, 18), 273 (8), 212 (16), 211 (100), 194 (7), 193 (38), 192 (14), 185 (9), 184 (9), 178 (30), 170 (12), 165 (3), 116 (10), 115 (91), 108 (8), 91 (25).

## 4. Conclusions

To summarize, an efficient and simple multicomponent reaction was developed for the preparation of 1,3-bis(sulfomethyl)imidazoliumate (bsmim), starting from the corresponding aminomethanesulfonic acid. Proper adjustment of the reaction conditions has allowed us to reduce the decomposition of the product (bsmim), in the reaction mixture, to the corresponding 1-methyl-3-sulfomethylimidazoliumate by elimination of sulfur(VI) oxide to negligible levels. Although the yield was moderate, the non-presence of the decomposition by-product facilitated the purification of the product. Despite the moderate reaction yield, it is proven that the multicomponent reaction has green metric values (i.e., AE, SF, MRP, RME, EcoScale, VMR, and E-factor), which makes this protocol acceptable-to-excellent in terms of sustainability and environmental impact. In addition, the catalytic activity of this second-generation IOS (bsmim) was tested in the Friedländer quinoline synthesis and the allylation of nitrogen-containing heterocycles, with the results obtained indicating that bsmim has much higher catalytic performance than its carboxy-imidazole derivatives. This type of ionic organic solid allows the promotion of reactions in the absence of solvent and mild reaction conditions, which improves the sustainability of organic synthetic processes. It is worth noting that the second-generation imidazolium IOS (bsmim) is really a progress compared with the bcmimCl, and a clear improvement compared with the analog bcmim.

## Data Availability

The data presented in this study are available in Appendix A.

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
