# Peer review of "Ionic Organic Solid 1,3-Bis(sulfomethyl)imidazoliumate as an Effective Metal-Free Catalyst for Sustainable Organic Syntheses"

_molecules, 2023, doi:10.3390/molecules28062695_

Round 1
Reviewer 1 Report
This manuscript by Pastor et al. presents an efficient preparation of a second-generation sulfo-imidazole acidic catalyst, 1,3-bis(sulfomethyl)imidazole (bsmim) , using a multicomponent reaction. The aim was to study its catalytic activity in the synthesis of quinoline derivatives, and in the allylic substitution of alcohols with nitrogen-containing heterocycles.
It is a very well written paper in standard English and is presented in an intelligible way .
The introduction explores the current state of the art and explains the aims of the work. The experimental design is well organized. The article structure, figures, tables and raw data shared are excellent.
 The manuscript also presents sufficient critical comparison to current established methods in terms of the "green-ness" of the methods.
In my opinion, the only observation that can be made regarding this manuscript concerns the possibility of recycling the catalyst. Have the authors tried to recycle and reuse the catalyst?
Once this is done, the paper deserves publication in this journal.
Author Response
We are grateful for the reviewer's positive assessment, considering that the work deserves to be published. Regarding your question:
Q: “Have the authors tried to recycle and reuse the catalyst?”
A: We agree that the recyclability is an interesting issue, especially in these catalysts. The tests we have performed have not given the desired results, due to the low thermal stability of the catalyst. As discussed in the optimization of bsmim synthesis, long reaction times, at high temperatures, end up degrading the IOS, albeit it keeps the activity during the reaction. To make this issue clearer, we have introduced the following comment in the manuscript: “As a final point, the stability of the bsmim (1) is compromised by prolonged heating, so the potential for recycling is reduced in the organic syntheses studied herein. Thus, in lieu of its higher activity, bsmim might be better suited for the design of milder synthetic processes, which would also better preserve its integrity.”
We hope that this revised manuscript answers all the concerns contained in the reviews. We feel that the revised manuscript is an improvement. We look forward to hearing from you regarding our submission. We would be glad to respond to any further questions and comments that you may have.
Sincerely yours,
Prof. Dr. Isidro M. Pastor
Reviewer 2 Report
The manuscript titled "Preparation of 1,3-bis(sulfomethyl)imidazole (bsmim) and its catalytic activity in organic synthesis" describes the effective synthesis of bsmim using a multicomponent reaction and its catalytic activity in two different reactions. The manuscript is well-written and easy to understand, with clear and concise language. The supporting information provided is adequate and helps to validate the experimental results.
The results presented in the manuscript are exciting and demonstrate the superior catalytic activity of bsmim over carboxy-imidazole analogs in the Friedländer reaction and allylic substitution reactions. The authors have provided a detailed analysis of the experimental results and discussed the underlying mechanisms behind the catalytic activity of bsmim. Overall, the results suggest that bsmim has great potential as a sustainable catalyst in organic synthesis.
As a reviewer, I fully support the publication of this manuscript. The work described is innovative and adds to the current knowledge of ionic organic solids in catalysis.
Some minor things needs to be adjusted.
For comparison it would be desirable to have the Lewis structures of bcmim and bcmimCl drawn out.
Put the VMR value obtained into context.
Author Response
We are grateful for the reviewer's positive assessment, supporting the publication of the manuscript. Regarding your question:
Q: “For comparison it would be desirable to have the Lewis structures of bcmim and bcmimCl drawn out.”
A: We agree that the structures of bcmim and bcmimCl may help in the reporting of this work. Thus, the structures have been included in the Table 2 (including the preparation of both).
Q: “Put the VMR value obtained into context.
A: We have considered this interesting comment. For that, we have calculated the green metrics for previous IOS (i.e. bcmim and bcmimCl) to compare. The data has been included in Table 2. In addition, we have included the following comment in the manuscript: “For comparison, green metrics for previous IOS (i.e. bcmim and bcmimCl) have been calculated (Table 2), obtaining VMR values similar to the calculated for bsmim. The slightly better VMR values for bcmim and bcmimCl are due to the higher reaction yields (almost double), which are related to the absence of decomposition of these IOS during their preparation.”
We hope that this revised manuscript answers all the concerns contained in the reviews. We feel that the revised manuscript is an improvement. We look forward to hearing from you regarding our submission. We would be glad to respond to any further questions and comments that you may have.
Sincerely yours,
Prof. Dr. Isidro M. Pastor
Reviewer 3 Report
Pastor and co-workers report the preparation of bsmim by MCR, employing aminomethanesulfonic acid, glyoxal and formaldehyde. The catalytic activity of the ionic solids has been tested in the construction of N-heterocyclic compounds.
The work represents a sustainable alternative and metal-free to the preparation of quinoline derivatives, performing reactions in the absence of solvent and mild reaction conditions. Thus I recommend that the manuscript is accepted in the present form. As I am not a English native speaker , my feeling is that some sentences could be improved.
Author Response
We are grateful for the reviewer's positive assessment, recommending the publication of the manuscript in the present form.
We hope that this revised manuscript answers all the concerns contained in the reviews. We feel that the revised manuscript is an improvement. We look forward to hearing from you regarding our submission. We would be glad to respond to any further questions and comments that you may have.
Sincerely yours,
Prof. Dr. Isidro M. Pastor